# A Cationic Contrast Agent in X-ray Imaging of Articular Cartilage: Pre-Clinical Evaluation of Diffusion and Attenuation Properties

**DOI:** 10.3390/diagnostics12092111

**Published:** 2022-08-31

**Authors:** Simone Fantoni, Ilenia Gabucci, Paolo Cardarelli, Gianfranco Paternò, Angelo Taibi, Virginia Cristofori, Claudio Trapella, Armando Bazzani, Marta Assenza, Alice Zanna Bonacorsi, Daniele Conti, Fabio Baruffaldi

**Affiliations:** 1Department of Physics and Astronomy, Alma Mater Studiorum, University of Bologna, 40127 Bologna, Italy; 2Istituto Nazionale di Fisica Nucleare (INFN), Division of Ferrara, 44122 Ferrara, Italy; 3Department of Physics and Earth Sciences, University of Ferrara, 44121 Ferrara, Italy; 4Department of Chemical, Pharmaceutical and Agricultural Sciences, University of Ferrara, 44121 Ferrara, Italy; 5Medical Technology Laboratory, IRCCS Istituto Ortopedico Rizzoli, 40136 Bologna, Italy

**Keywords:** X-ray imaging, articular cartilage, contrast agent, contrast-enhanced computed tomography, proteoglycans, post-traumatic osteoarthritis

## Abstract

The aim of this study was the preliminary assessment of a new cationic contrast agent, the CA4+, via the analysis of spatial distribution in cartilage of ex vivo bovine samples, at micrometer and millimeter scale. Osteochondral plugs (*n* = 18) extracted from bovine stifle joints (*n* = 2) were immersed in CA4+ solution up to 26 h. Planar images were acquired at different time points, using a microCT apparatus. The CA4+ distribution in cartilage and saturation time were evaluated. Tibial plates from bovine stifle joints (*n* = 3) were imaged with CT, before and after 24 h-CA4+ bath immersion, at different concentrations. Afterward, potential CA4+ washout from cartilage was investigated. From microCT acquisitions, the CA4+ distribution differentiated into three distinct layers inside the cartilage, reflecting the spatial distribution of proteoglycans. After 24 h of diffusion, the iodine concentration reached in cartilage was approximately seven times that of the CA4+ bath. The resulting saturation time was 1.9 ± 0.9 h and 2.6 ± 2.9 h for femoral and tibial samples, respectively. Analysis of clinical CT acquisitions confirmed overall contrast enhancement of cartilage after 24 h immersion, observed for each CA4+ concentration. Distinct contrast enhancement was reached in different cartilage regions, depending on tissue’s local features. Incomplete but remarkable washout of cartilage was observed. CA4+ significantly improved cartilage visualization and its qualitative analysis.

## 1. Introduction

Articular cartilage is an avascular and aneural tissue organized as a hierarchical structure which enables nearly frictionless motion due to the interplay between its constituents. In particular, the aqueous phase, formed by the interstitial fluid, and the extracellular matrix (namely collagen, proteoglycans and chondrocytes), confer excellent mechanical properties, comprising lubrication, compressive and shear stiffness [1].

Traumas and prolonged load exertion on joints damage the extracellular matrix, leading to wear of collagen fibers and loss of proteoglycans. These conditions negatively impact mechanical properties of articular cartilage, resulting in the development of post-traumatic osteoarthritis (PTOA) [2]. The diagnosis of PTOA occurs when the disease is already in its late stages, including pain and movement impairment, and no pharmacological treatment has yet been effective [3]. Thus, early detection of PTOA is crucial.

In clinical routine, magnetic resonance imaging (MRI) and computed tomography (CT) play a major role in assessing the state of articular cartilage [4,5]. Based on nonionizing radiation, MRI shows excellent contrast of articular cartilage and is able to provide quantitative information [6], despite its long acquisition times and its limited spatial resolution [7]. However, MRI delivers poor information about underlying subchondral bone condition, individuated as an additional indicator of PTOA development [8]. On the other hand, CT offers high spatial resolution and provides excellent imaging of subchondral bone, imparting low doses to patients [9]. The critical shortcoming of X-ray imaging lies in low distinguishability of soft tissues, owing to their similar X-ray attenuation properties.

In the frame of preclinical studies, histology and biochemical assay are the golden standard for quantitative assessment of cartilage state [10], despite their destructiveness and time-consuming procedures. Additionally, histology provides poor information on volume features.

Essential improvement of cartilage visualization can be achieved with the use of contrast agents [11]. Previous studies proved the feasibility of contrast enhancement of cartilage with contrast agents currently used in clinical routine [12], carrying out quantitative analysis and correlation to independent measurements (i.e., biomechanical tests, histological assessments, MRI acquisitions) [13].

Recently, attention was given to targeted cartilage constituents, such as proteoglycans, identified as potential biomarkers for PTOA diagnosis and stage classification [14,15]. A novel cationic iodinated contrast agent, CA4+, was specifically designed to electrostatically interact with glycosaminoglycans (GAGs) bound in proteoglycans [16]. In particular, the Coulomb attraction between negative fixed charges of GAGs and positively-charged CA4+ molecules showed superior contrast enhancement compared to commercially available contrast agents, (even at lower concentrations [17,18,19]), in animal [20,21,22,23] and human [24,25,26,27,28,29] samples, on several types of tissue containing GAGs [13,30,31] and cartilage tissue-engineered constructs [32]. The use of CA4+ was investigated in combination with synchrotron radiation as well, exploiting the high brilliance and monochromaticity of X-ray beam [33,34,35,36].

The aim of this study was the preliminary assessment of CA4+ spatial distribution in ex vivo bovine samples, at different scales. In particular, the depth-wise distribution of CA4+ in cartilage at different time points was analyzed and the time required for cartilage to reach an attenuation value equal to 95% of bath attenuation was evaluated at micrometer scale using a microCT apparatus on small-sized bovine osteochondral samples. Additionally, the three-dimensional spatial distribution of CA4+ in cartilage was assessed at millimeter scale using a clinical CT apparatus on bovine tibial plates.

MicroCT analysis provided information about the diffusion times and the depth-wise distribution of CA4+ at micrometer scale. The saturation time was then used to validate the times considered for the subsequent immersion of tibiae in CA4+ solution for clinical CT acquisitions. Clinical CT acquisitions highlighted the outcome of CA4+ on tibial plates, delivered with instrumentation regularly employed in clinical routine. In particular, the resulting three-dimensional iodine map at millimetric scale pointed out the differential CA4+ distribution, which in turn depended on cartilage’s local features. Since the animal model chosen for the experiment (namely, the bovine model) approximated the human tibial plate very well, the clinical CT measurements could anticipate the possible CA4+ outcome on human knee condyles.

## 2. Materials and Methods

### 2.1. MicroCT Acquisitions

#### 2.1.1. Samples Preparation

Osteochondral plugs (*n* = 18) were extracted from tibial plates and femoral condyles of distal ends, from bovine stifle joints (*n* = 2). All bovine joints used for this study were obtained from a local slaughterhouse for human nutrition, within 24 h of slaughtering. No animals were sacrificed for the aims of this study. The osteochondral plugs were harvested from an accurately prepared articulating cartilage using a computer numerical control machine equipped with a 10-mm diameter diamond-tipped cylindric cutter irrigated with saline solution at room temperature. From the extraction procedure, osteochondral plugs resulted in 10-mm diameter and variable height (2–3 cm in average), each including articular cartilage, subchondral and trabecular bone. After harvesting, the osteochondral plugs were enveloped in phosphate-buffered saline (PBS)-soaked lint and frozen at T = −20 °C until image acquisition.

#### 2.1.2. Contrast Agent Synthesis and Solution Preparation

5,5′-[Malonylbis(azanediyl)]bis[N1,N3-bis(2-aminoethyl)-2,4,6-triiodoisophthalamide] chloride salt (CA4+) was the contrast agent used in the present study. It is a hexaiodinated molecule characterized by a positive net +4 charge, a molecular weight 1354 g/mol and an iodine content fraction per molecule equal to 0.5084. It was synthetized according to the work of Stewart and colleagues, with minor modifications [18]. In particular, the last step of acidic Boc-deprotection of amine was carried out using a solution of HCl in dioxane instead of trifluoroacetic acid, in order to afford the desired tetrakis chloride salt and avoid ion exchange.

The CA4+ solutions, at various iodine concentrations, were obtained by dissolving CA4+ salt in deionized water, and balancing to pH = 7.4 by the addition of a NaOH 4-M solution.

#### 2.1.3. Calibration Curve Evaluation

Prior to the image acquisition of osteochondral plugs, the attenuation values of CA4+ solutions at different concentrations were assessed (namely 0, 1, 2, 5, 10, 20 mgI/mL). All solutions were obtained by adding CA4+ salt to deionized water. Each solution was imaged once poured in a polymethyl methacrylate (PMMA) custom-made sample holder. The attenuation for each solution was evaluated as the mean value of a rectangular ROI, drawn in the image center. The resulting calibration curve was used in a second moment for conversion of gray levels into effective iodine concentrations reached within the cartilage.

#### 2.1.4. Osteochondral Plugs Acquisition

Planar images were acquired using a microCT apparatus (SkyScan 1072, SkyScan, Aartselaar, Belgium), recording one-dimensional profiles along the contrast medium diffusion direction. Each osteochondral plug was thawed in PBS and inserted in the sample holder, leaving only the cartilage exposed.

The sample-holder featured an external diameter of 12 mm and an inner diameter of 10 mm. Therefore, the osteochondral plug could easily be inserted. The container could host osteochondral plugs with variable height and the contrast agent bath at the top the cartilage surface. Provided that the height of the sample-holder interior was equal to approximately 3 cm, a total of 3 mL of volume was admitted inside the sample-holder, including the osteochondral plug and the CA4+ bath.

Prior to the insertion of the osteochondral plugs in the sample-holder, the inner walls of the latter were smeared with silicone-based grease to prevent any leakage of CA4+ through the lateral edges of articular cartilage during the immersion in the contrast agent bath. Since the inner diameter of the sample-holder approached the diameter of the osteochondral plugs, the amount of silicone-based grease used was minimal.

The baseline images were acquired immersing the plugs in PBS. Afterward, PBS was removed and a fixed volume of CA4+ solution (0.5 mL) was poured above the cartilage surface. Planar images were acquired at different time points, up to 26 h (namely 2, 5, 10, 15, 20, 30, 45 min, 1, 1.5, 2, 3, 4, 5, 6, 24, 25, 26 h), leaving the samples in CA4+ bath at room temperature. The amount of CA4+ solution was chosen according to the cartilage thickness. In particular, considering cartilage thicknesses corresponding to 2 mm, the average cartilage volume would have been 0.157 mL. As a result, the CA4+ volume chosen for the microCT experiment was approximately three times the cartilage volume.

MicroCT parameters were kept fixed for both solution and osteochondral plugs acquisitions (X-ray tube voltage = 50 kV, current = 197 µA, 1-cm Al filter, pixel size = 11.52 µm, exposure time = 5.9 s).

#### 2.1.5. Data Elaboration

Data elaboration was performed using a custom-made Matlab (MATLAB 2021a, MathWorks, Natick, MA, USA) code. In order to evaluate the concentration of CA4+ dispersed in cartilage, the baseline image was subtracted to the images of samples exposed to CA4+ solution. The procedure allowed us to single out the attenuation due to CA4+ only in both the cartilage and contrast medium bath. The attenuation value reached in cartilage was evaluated as the mean value of a rectangular ROI in each subtracted image, drawn in the image center. The attenuation against immersion time in CA4+ bath was fitted according to Fick’s second law,
(1)$$A=a\left(1-e^{\frac{t}{\tau }}\right)$$
given τ the characteristic diffusion time of CA4+ within the cartilage, by definition the time it took the cartilage to reach 65% of maximum attenuation in CA4+ bath. Hence, the saturation time τ_95_, namely the time required for the cartilage to reach 95% of attenuation with respect to CA4+ solution bath, was evaluated for both tibial and femoral samples, considering the following formula
$$\tau_{95}=5\tau .$$

The saturation time was used to validate the immersion time of tibiae in CA4+ bath for te clinical CT acquisitions.

The conversion from attenuation value to iodine concentration was performed for both cartilage and CA4+ bath, on ROI extracted from subtracted images. The ROI was 100 pixels wide (equivalently, 1.15 mm), while its height was adjusted to cartilage thickness. Afterward, the ratio of iodine concentration in cartilage to the one of CA4+ solution bath was computed.

#### 2.1.6. Statistical Test

The one-sample Kolmogorov–Smirnov test was used to check the normal nature of data distribution, with 5% significance level.

### 2.2. Clinical CT Acquisitions

After diffusion time was investigated from micrographs, millimeter-scale assessments were performed using a clinical computed tomography apparatus (Revolution^TM^ CT, GE Healthcare, Waukesha, WI, USA), on distal-end tibiae of bovine stifle joints (*n* = 3).

The used CT apparatus relied upon fast X-ray tube voltage switching technology and was capable of acquiring multiple images simultaneously at two different tube voltages (80/140 kVp, slice thickness 0.625 mm) in single source-detector configuration. Additionally, virtual monochromatic reconstruction images ranging from 40 to 140 keV could be obtained, exploiting the Gemstone Spectral Imaging (GSI) Xtream image generation flow [37].

#### 2.2.1. Phantom Acquisition

A phantom containing five flasks, including CA4+ solutions with different iodine concentrations (0, 5, 10, 15, 20 mgI/mL), was imaged. The phantom reconstructions were carried out for further data elaboration.

#### 2.2.2. Samples’ Preparation and Acquisition

After careful disarticulation of bovine joints and removal of unnecessary tissues, *n* = 3 tibial plates were obtained. The lateral size of the tibial plates was 15 cm in average, while the bone section was approximately 25 cm long. Four markers for image registration (PinPoint Multi-Modality Fiducial Markers, BeekleyMedical, Bristol, CT, USA) were applied using cyanoacrylate on bone surface of each tibia, once properly smoothed. Customized sheaths were realized by modeling bicomponent silicone on tibiae surface in order to reduce the required volume of contrast medium. Afterwards, PBS-imbued lint was applied on articulating surfaces, and tibiae were frozen (T = −20 °C).

Prior to CA4+ immersion, the tibiae were thawed for 24 h in PBS bath at 6 °C and baseline scan of the volumes were acquired. Hence, each tibia was immersed in CA4+ bath (100 mL) at a single concentration, among 5, 10, 15 mgI/mL, up to 24 h at 6 °C and imaged. The immersion time was selected according to the saturation time resulting from the microCT acquisitions. The selection of CA4+ concentrations was based on evidence in the work of Karhula et al. [24]. Virtual monochromatic reconstructions at 70 keV were obtained, as the selected energy guaranteed an adequate differentiation of several tissues in samples.

Following these acquisitions, CA4+ washout from cartilage was investigated. Tibiae were immersed in PBS bath (300 mL) at 6 °C for 24 h and imaged. After the PBS solution was changed, the same procedure was repeated, in order to reach 48-h washout.

#### 2.2.3. Data Elaboration

The calibration curve was carried out from CT acquisitions of CA4+ flasks at different concentrations (0, 5, 10, 15, 20 mgI/mL), scanned and virtually reconstructed at 70 keV. The calibration curves allowed the subsequent conversion of HU to iodine concentration in reconstructed volumes of cartilage. Overall attenuation was evaluated for each CA4+ concentration bath, before and after the immersion of samples in the contrast medium solution. Three different tissues were identified and analyzed, namely cartilage, bone and connective tissue. The CT numbers were evaluated from 6 ROIs, taken from 4 subsequent slices, for a total of 24 ROIs per each tissue type, so as to carry out a statistical analysis. The ROIs were circular, of 5 pixels-radius (alternatively, 1.6-mm radius, provided the pixel size of 0.32 mm).

Acquisitions before and after CA4+ bath were coregistered and subtracted, in order to obtain cartilage volumes only, and contrast enhancement was evaluated. Further manual segmentation was performed to exclude spurious pixels (namely, artifacts resulting from coregistration and subtraction of images). As a result, cartilage CT numbers from segmented volumes would be attributable to iodine content only. The CT numbers were extracted from 18 volumes of interest (VOI), in different cartilage regions, for each CA4+ iodine concentration. The VOIs considered were cylindrical, with a diameter approximately equal to the one of osteochondral plugs examined in the microCT analysis.

Eventually, the iodine concentration dispersed in cartilage and the segmented volumes were evaluated as a function of time in order to investigate potential removal of CA4+ from cartilage following immersion in PBS solution. The washout behavior was fitted with the following function:(2)$$c=a_{w}e^{-b_{w}t}\;$$
and the washout time was evaluated as the time required by cartilage to reach 5% attenuation, with respect to the attenuation value assumed after 24 h immersion in CA4+ solution.

#### 2.2.4. Statistical Test

To evaluate the distinguishability among tissues, the nonparametric two-samples Mann–Whitney test and two-samples Kolmogorov–Smirnov test, both with 5% significance level, were considered. Both tests determine if two samples belong to the same continuous distribution. In particular, the Mann–Whitney test determines whether one distribution is stochastically greater than the other. The Kolmogorov–Smirnov test response is based on differences computed on distances between the empirical cumulative distribution functions. Possible overlap of histograms was further evaluated, and its significance discussed using Cohen’s d metric.

The adoption of 5%-significance level Kolmogorov–Smirnov tests verified the normal distribution of cartilage CT numbers, evaluated from the selected VOIs.

## 3. Results

### 3.1. MicroCT Acquisitions

#### 3.1.1. Calibration Curve

Calibration data are reported in Figure 1 and were fitted to the linear function *y = a + b·x*, relating the iodine concentration to CA4+ attenuation. The parameters of the calibration curve, a = 0.7531 GL and b = 1.6189 GLml/mgI (R^2^ = 0.9955), were necessary for the conversion of gray levels into iodine concentrations.

#### 3.1.2. Osteochondral Plugs

From a visual inspection, it was possible to observe the effect of CA4+ diffusion on cartilage visibility, as displayed in Figure 2.

For instance, in the microradiograph reported in Figure 2b, it was possible to sharply distinguish the articular cartilage from the CA4+ bath, after 24 h immersion. Contrariwise, observing Figure 2a, in the planar image of the same sample, it was not possible to recognize the cartilage tissue from the PBS bath, due to their similar X-ray attenuation properties.

For each sample, nearly the same diffusion process was observed and three distinct layers were recognizable, as displayed by attenuation profiles in Figure 3. In particular, at the first time points, the superficial layer showed augmented attenuation. For longer exposure times, the mid-deep layer progressively grew more radiopaque. Eventually, nearly the same attenuation was observed in cartilage between 6- and 24-h time points for both femoral and tibial samples, as reported in Figure 3.

The saturation times, namely, the time required for the cartilage to reach 95% of bath attenuation, were τ^F^_95_ = 1.9 ± 0.9 h and τ^T^_95_ = 2.6 ± 2.0 h for femoral and tibial samples, respectively. The iodine concentration reached in the cartilage was computed using the calibration curve previously obtained. Such conversion was carried out after baseline images were subtracted to each time point acquisition after CA4+ immersion, in order to exclude any attenuation attributable to water and PMMA. The diffusion curves, obtained by collecting the average values of attenuation and iodine concentration as a function of time, are reported in Figure 4.

The average values were computed on a rectangular ROI centered in the middle of the cartilage, as described in Material and Methods. The iodine concentrations reached in cartilage at 24 h were c^T^_24_ = 19.5 ± 5.5 mgI/mL and c^F^_24_ = 21.2 ± 4.9 mgI/mL for tibial and femoral samples, respectively. Alternatively, the relative iodine concentrations with respect to CA4+ bath could be evaluated, provided the bath attenuation after 24 h immersion. In particular, the relative iodine concentrations achieved were c^F,rel^_24_ = 8.3 ± 2.4 and c^T,rel^_24_ = 6.2 ± 3.5 for femoral and tibial samples, respectively.

### 3.2. Clinical CT Acquisitions

#### 3.2.1. Calibration Curve

The calibration curve, shown in Figure 5, was carried out from CT scans of CA4+ flasks at different concentrations (0, 5, 10, 15, 20 mgI/mL). It enabled the conversion of cartilage attenuation, observed in subtracted volumes, to the effective iodine concentration reached within cartilage. A linear model was assumed, and parameters for following calibration were a = −5.589 HU and b = 23.537 HUml/mgI (R = 0.9969).

#### 3.2.2. Attenuation Comparison

For each tibia, attenuation was compared among three different types of tissue, namely connective, bone and cartilage. A total of 24 ROIs were taken per tissue type, and distribution of pixel attenuation was reported in histograms, as shown in Figure 6. This study was carried out for tibiae prior and after CA4+ immersion. Results from statistical analysis, including Cohen’s d, two-samples Kolmogorov–Smirnov test and Mann–Whitney test, are reported in Table 1 and Table 2 for tibiae prior to and after CA4+ immersion, respectively.

#### 3.2.3. Iodine Concentration within Cartilage

The iodine concentration reached in cartilage was evaluated, converting the attenuation values through the calibration curve obtained from phantom CT reconstruction. The mean concentration values were computed from the 18 VOIs evaluated for each tibia. In Figure 7a, the iodine concentration is plotted against time. The washout behavior was fitted to the following model:(3)$$c=a_{w}e^{-b_{w}t}$$
returning R_5_^2^ = 0.7822, R_10_^2^ = 0.8836, R_15_^2^ = 0.9715 for 5, 10 and 15 mgI/mL-concentration curves, respectively.

The washout times, namely, the time required by cartilage to reach the 5% of attenuation due to CA4+, were t^w^_5_ = 301 ± 41 h, t^w^_10_ = 233 ± 20 h and t^w^_15_ = 245 ± 22 h for 5, 10 and 15 mgI/mL-concentration, respectively.

### 3.3. Cartilage Volume and Thickness Estimation

The cartilage volume was estimated, summing up all pixels included in segmented volumes. The estimated volumes of articular cartilage were 6.38 ± 0.01 cm^3^, 8.13 ± 0.08 cm^3^ and 6.87 ± 0.01 cm^3^ for 5, 10 and 15 mgI/mL, respectively.

Cartilage thickness was calculated as mean value of transversal number of pixels, computed from each VOI. The resulting values were 1.8 ± 0.2 mm, 2.8 ± 0.1 mm and 2.3 ± 0.1 mm for 5 mgI/mL, 10 mgI/mL and 15 mgI/mL-concentrations, respectively. As displayed in Figure 7b, the tibia immersed in 10 mgI/mL-bath showed the largest cartilage thickness, probably due to the immersion procedure in silicon sheath.

## 4. Discussion

In X-ray imaging, without contrast enhancement, cartilage is merely indistinguishable with respect to surrounding soft tissues, owing to similar attenuation properties. Therefore, any attempt to evaluate cartilage main features is limited to rough morphometric assessments (i.e., intra-articular space).

In this preliminary work, we showed that the use of a high-affinity cationic contrast agent considerably enhanced cartilage visibility, exploiting a microCT system and a clinical CT system, in bovine samples of different size. As shown by previous works [20], the electrostatic attraction of CA4+ to the negative fixed charge of proteoglycans, a functional constituent serving as potential biomarker of cartilage state, conferred radiopacity to specific cartilage structures.

From previous studies, no evidence of complete tibial condyle acquisition with clinical CT emerged. If studies including intact joints were considered, assessments were carried out with in vivo microCT, as the animal model used (e.g., rabbit, rat) could not be investigated with any clinical CT system. Furthermore, those models did not replicate the size of the human knee joint.

The only work found that focused on whole stifle joint was from Stewart et al. [23]. In particular, it investigated the equine femoropatellar joint and how the CA4+ injection highlighted cartilage defects, using a clinical CT system. However, no attention was paid to tibial cartilage, though its deterioration, together with the femoral cartilage, was implied by the development of osteoarthritis.

The present work aimed to fill the gap by considering the outcome of CA4+-induced contrast enhancement on whole tibial plates, by means of a clinical CT system. Furthermore, the present study focused on the positive impact of CA4+ use on differentiation between three different tissues (namely, connective tissue, cartilage and bone), clinically relevant for evaluation of joint states.

The calibration curves in Figure 1 and Figure 5 displayed a linear behavior, as seen from other works. For instance, by comparing the curve in Figure 5 of the present work, and the curve in Figure 2b from the work of Bhattarai at al. [25], both curves were definitely linear. However, it was not possible to carry out further comparison as the units and scales displayed on y-axis of the curve of Bhattarai et al. were different from the present work.

As a matter of fact, microradiographs showed a clear distinction of three layers within cartilage, namely superficial, mid and deep layer, depending on different attenuation, found as a function of cartilage depth. Independent methods for cartilage quantitative analysis pointed out this hierarchical structure, as confirmed by histology [38], microspectrophotometric analysis [39], magnetic resonance [40] and previous works adopting contrast-enhanced microCT [15,17,18,19,22,41]. In the present study, microCT acquisitions provided information on diffusion kinematics. The visualization of the superficial layer was already enhanced after short diffusion times, namely within one hour. For longer diffusion times, CA4+ progressively accumulated in the mid layer, until the same attenuation was reached down to the deep zone. Within 24 h, the deep layer could be recognized as the most radiopaque. Nearly the same diffusion as iws observed for both tibial and femoral samples. Thus, it could be inferred that CA4+ drifted toward deeper layers, supposedly owing to the gradient of proteoglycans concentration. This fact confirmed the high affinity of CA4+ to negatively-charged proteoglycans.

The net iodine concentration within cartilage showed nearly the same values for both femoral and tibial samples. Furthermore, cartilage reached higher iodine concentrations if compared to initial CA4+ bath, as shown in Figure 4, rendering it clearly distinguishable.

Analyzing the CA4+ diffusion curves, nearly the same attenuation value was reached in cartilage comparing the 6-h and the 24-h time points, as diffusion reached equilibrium. In particular, the saturation time points out that diffusion equilibrium was already reached before the 24-h time point by both tibial and femoral samples. Furthermore, the presented saturation time was consistent with those carried out by previous works, despite the different acquisition modality [17], and it could be considered for further experimental studies with CA4+. Eventually, the relative iodine concentrations within cartilage of femoral osteochondral plugs were consistent with those assessed from previous studies performed on ex vivo bovine samples [20].

The fluctuations associated to the attenuation values shown in the diffusion curve were attributable to the heterogeneous local properties of cartilage. In particular, morphological (i.e., thickness) and compositional (i.e., proteoglycans content) features can be expected to vary, depending on the extraction site, on articulating cartilage [42]. Such fluctuations, highlighted by the error bars in Figure 4, impacted the parameters of the fitting function, and, in turn, the evaluation of the saturation time. As a result, a discrepancy of 42 min was found between the saturation time determined for femoral and tibial osteochondral plugs. On the other hand, the magnitude of the fluctuations found in attenuation values (referring to error bars in Figure 4) depended on the number of samples included in the study. A greater number of samples should settle such fluctuations to lower values.

These remarks were confirmed by clinical CT assessments. As a matter of fact, the VOIs provided different cartilage thicknesses, as reported in Figure 6, with fluctuations from mean values ranging from 5% to 10%. Different thickness values were attributable to variations of CA4+ permeation in cartilage.

Furthermore, the immersion operation of the tibia distal end in silicon sheath was user-dependent.

The iodine concentration within cartilage may reflect the heterogenous distribution of proteoglycans in extracellular matrix, and supporting evidence might be provided by independent measurements, namely biomechanical indentation [43].

As shown in Figure 6, it was possible to clearly distinguish the cartilage from other connective tissues following the CA4+ bath. Contrarily, soft tissues could not be discriminated prior to CA4+ exposure, as they shared nearly the same attenuation properties. Furthermore, the simultaneous assessment of cartilage and the underlying subchondral bone state could be feasible if advanced imaging techniques (i.e., dual-energy) were adopted [44]. In particular, an increased shift of cartilage attenuation toward higher values was found for increasing CA4+ bath concentration. Evidence of the trend can also be found in the work of Karhula et al. [24], as was possible to observe from whole sample data in Figure 3 of the same work, though the samples and CA4+ concentration were different from the present study. In particular, the work of Karhula et al. included human osteochondral plugs as samples (2 mm-diameter, *n* = 9 osteochondral plugs from human femurs, 3 for each iodine concentration), and 3, 6, and 24 mgI/mL as CA4+ solution concentrations.

The washout baths led to the partial removal of CA4+ from cartilage. Previous works pointed out different findings, even if the same animal model (namely, bovine) were adopted. Works of Bansal et al. [17,19] demonstrated that, after 24-h PBS washout, a 50% decrease of CA4+ content was observed from cartilage of osteochondral plugs harvested from femoral distal ends. In the present study, the expected time for complete cartilage washout was supposedly hindered by the relatively low temperature (T = 6 °C).

The choice of acquiring fast planar images avoided any bias introduced by the ongoing diffusion process of CA4+ within cartilage, potentially occurring during tomographic acquisitions. On the other hand, only 2-dimensional information was provided. Due to the heterogeneous nature of samples, image projections could have included tissue regions with different local properties. Image acquisition should avoid any overlap between cartilage and subchondral bone, cartilage and CA4+ bath. As a consequence of having chosen a planar imaging modality, it was therefore not possible to distinguish any calcified cartilage at the interface between subchondral bone and the deep layer of articular cartilage. Furthermore, planar acquisitions are less accurate than three-dimensional scans. For instance, depth-wise profiles provided in the work of Saukko et al. [33] better highlighted the three cartilage layers.

Moreover, planar acquisitions were analyzed assuming that sample thickness was constant. In order to satisfy this assumption, ROIs for attenuation evaluation were taken from plug centers, acknowledging the cylindrical geometry of samples. Nevertheless, the iodine concentration considered for the micrometer acquisitions did not give rise to any substantial beam hardening, as was possible to observe from the calibration curve in Figure 1.

It is noteworthy that there as accordance between the iodine concentration in tibial osteochondral plugs (measured with microCT) and tibiae (measured with the clinical CT), reached after 24-h CA4+ immersion at 10 mgI/mL concentration. As a matter of fact, microCT assessment pointed out that nearly the same attenuation was reached between 6 h and 24 h. Hence, the bath time adopted for tibiae was adequate, though lower temperatures ruled out CA4+ diffusion. Furthermore, the different sample preparation adopted for microCT and clinical CT acquisitions, respectively, seemingly did not impact on CA4+ final uptake. The washout investigation was based on preliminary results, as a limited number of experimental points (namely 3) were included.

The sample preparation for the present work did not include the use of any agent preventing cartilage degradation (i.e., antimycotic, antibiotic, protease inhibitors), unlike previous studies [17,18,20,21,23,24,25,26,27,28,29,33,35,36]. Though deterioration processes are expected to affect cartilage following the harvesting of samples, the use of preserving agents could alter cartilage pristine properties, including the interaction with CA4+. Furthermore, freeze-thaw cycles must be accounted for in potential modification of cartilage original properties, i.e., integrity of extracellular matrix [45,46]. On the other hand, several studies showed that one freeze-thaw cycle, as was the case of the present work, is compatible with biochemical and biomechanical assessments [47].

This study presented several limitations. A limited number of samples (namely *n* = 18 and *n* = 3 for the microCT and clinical CT acquisitions, respectively) were considered for this work. As a consequence, specifically for the clinical CT assessment, values of cartilage volume and thickness showed fluctuations. A larger number of samples could reduce the intra-specimen discrepancies. An additional source of bias might be the procedure of manual segmentation, performed in order to remove spurious pixels and artifacts rising from volume subtraction. Eventually, the realization of custom silicon sheathing for each tibia could potentially impact the CA4+ bath and, in turn, influence the CA4+ uptake in cartilage.

MicroCT and clinical CT acquisitions were carried out under different conditions (i.e., bath temperature). Lower temperatures were chosen for the clinical CT acquisitions in order to slow down the natural deterioration of samples. That choice slowed down the diffusion process of CA4+ towards cartilage of tibiae. The same consideration was true for the washout procedure, as the resulting washout times scales were larger than the uptake ones. Contrarily, microCT acquisitions were conducted at room temperature, since no control of such parameter was possible with the experimental setup.

Another shortcoming of the present work was the lack of control over the osmolality of the obtained CA4+ solution. Any difference in osmolality between the internal and external cartilage environment could havee affecdt diffusion and morphological properties, though slight modifications were observed on equine samples with nonionic small molecule contrast agent [48]. Conversely, the use of hyperosmolar contrast agents heavily impacted cartilage mechanical properties, causing tissue softening and shrinkage [49]. As a consequence, possible effects of CA4+ solution osmolality should be taken into account in the frame of conjoint biomechanical investigations.

The environmental conditions were far from the physiological ones. The adoption of osteochondral plugs for microCT investigation removed the assumption of continuity of cartilage extracellular matrix. The preparation of tibiae included the disruption of local joint environments, such as the synovial capsule. Processes adjuvating CA4+ diffusion were not considered, namely pressurization, mechanical agitation of CA4+ on cartilage surface or the adoption of motion protocols. The aforementioned procedures might positively impact uptake time of CA4+, and further reduction of CA4+ uptake times could have interesting applications in in vivo preclinical tests [50], provided the promising results from assessments of CA4+ safety, pharmacokinetics, and excretion on animal model [18].

## 5. Conclusions

This study confirmed the effective contrast enhancement of articular cartilage induced by a high-affinity cationic CA4+ contrast agent. Different approaches unveiled the potential of CA4+ for cartilage visualization on different scales. From micrometric analysis, the uptake times revealed by diffusion study were valuable premises for further investigation and in vivo preclinical trials. The spatial distribution of CA4+ was coherent to the depth-wise concentration of proteoglycans, found in healthy articular cartilage. Even lower iodine concentrations allowed significant enhancement of cartilage contrast, implying a reduced onset of possible side effects in in vivo experiments. The targeted interaction with proteoglycans might be exploited as a diagnostic tool for PTOA identification.

The analysis of clinical CT acquisitions confirmed the enhancement of cartilage visualization, with iodine concentrations fluctuating according to local features of articular cartilage. The preliminary results of PBS washout demonstrated partial CA4+ removal from articular cartilage.

The simultaneous assessment of cartilage and subchondral bone state, provided by microCT, could shed light on the interplay of possible factors triggering PTOA. In particular, the adoption of advanced imaging techniques, namely dual-energy, could provide unvaluable morphological and quantitative information.

Assuming the compatibility of bovine model to human in osteoarthritis studies [51], it could be possible to replicate the microCT study on ex vivo human osteochondral plugs. Recent microCT systems, featuring high resolution acquisitions and reduced scan times, should allow three-dimensional assessments besides planar acquisitions.

Obviously, ex vivo experiments are destructive, as the extraction of samples requires explants from cadavers or arthroplasties. Samples must be suitable for the microCT system; thus their size tends to be small. Furthermore, samples harvested from arthroplasties’ explants might already show development of osteoarthritis, even at late stages. On the other hand, those drawbacks could be avoided if pre-clinical microCT systems were adopted.

In particular, pre-clinical microCT systems could perform high resolution peripheral quantitative CT (HRpQCT) in vivo acquisitions on distal ends of patients’ limbs [52,53,54]. It would thus be possible to carry out in vivo assessment, and the dose deposited to patients would be intrinsically reduced compared to conventional CT, as reported by recent literature [9]. Provided approval by the FDA is achieved and the design of an optimal protocol for contrast agent usage in patients is carried out, the application of contrast agents should enable the contrast enhancement of articular cartilage. As a consequence, the state of both articular cartilage and subchondral bone would be unveiled, and possible staging of osteoarthritis would be performed.

Previous studies proved the correlation between contrast-enhanced CT (CECT) measures and other independent techniques. Reference methods included histological assessments, renowned for their destructiveness and lack of three-dimensionality, and MRI acquisitions, recognized for high cartilage sensitivity but hindered by poor spatial resolution. Aside from its diagnostic potentialities, significant correlation to biomechanical measures supports CA4+ CECT as a valuable tool for cartilage characterization.

Further activities will be directed toward the confirmation of the aforementioned evidence.

## Figures and Tables

**Figure 1 diagnostics-12-02111-f001:**
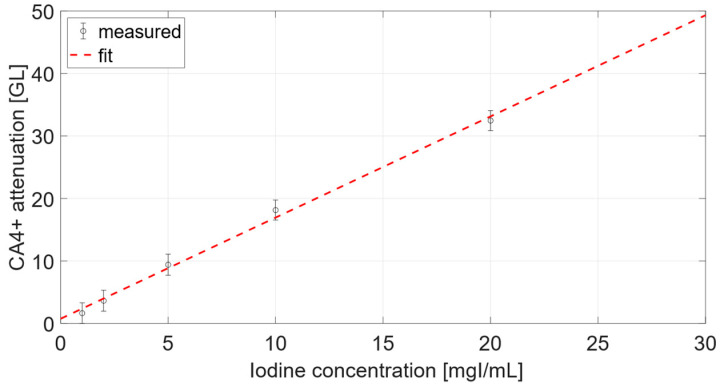
Calibration curve relating attenuation and iodine concentration of CA4+ from microCT radiographs.

**Figure 2 diagnostics-12-02111-f002:**
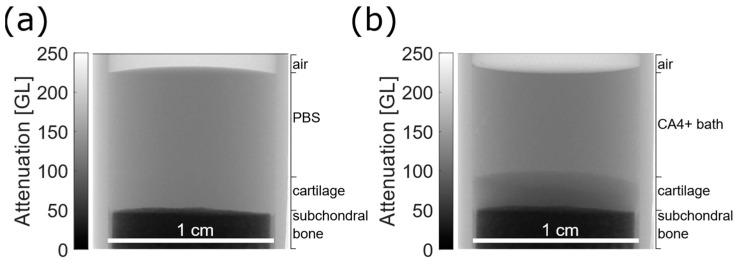
Results from microradiographs. (**a**) Microradiograph of osteochondral plug in PBS bath, prior to CA4+ immersion. Lateral edges are the walls of PMMA sample holder. (**b**) Microradiograph of the same osteochondral plug after 24 h immersion.

**Figure 3 diagnostics-12-02111-f003:**
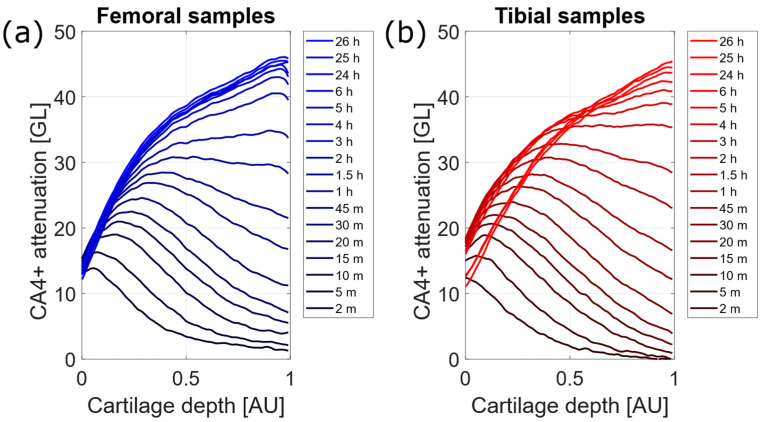
Mean attenuation profiles reported as a function of cartilage depth for femoral (**a**) and tibial (**b**) samples. Each attenuation profile refers to a specific diffusion time, as pointed out by the legend. Cartilage depth is drawn from cartilage surface (namely 0) to cartilage tidemark (namely the cartilage-subchondral bone interface, 1).

**Figure 4 diagnostics-12-02111-f004:**
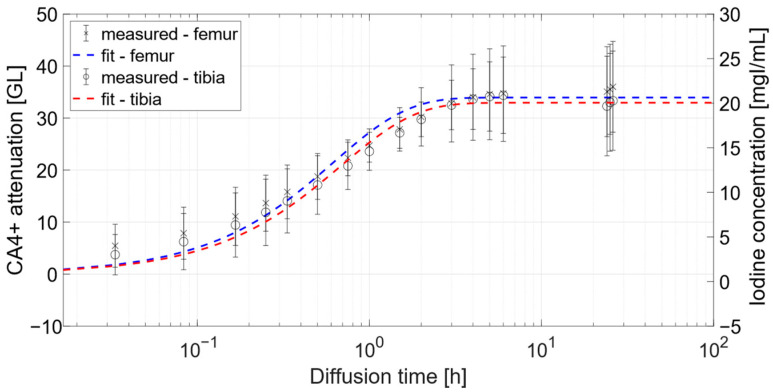
Diffusion curve of CA4+ in femoral (blue) and tibial (red) osteochondral plugs, pointing out attenuation (on left axis) and iodine concentration (on right axis) reached in cartilage, with respect to immersion time, in 10 mgI/mL-concentration bath.

**Figure 5 diagnostics-12-02111-f005:**
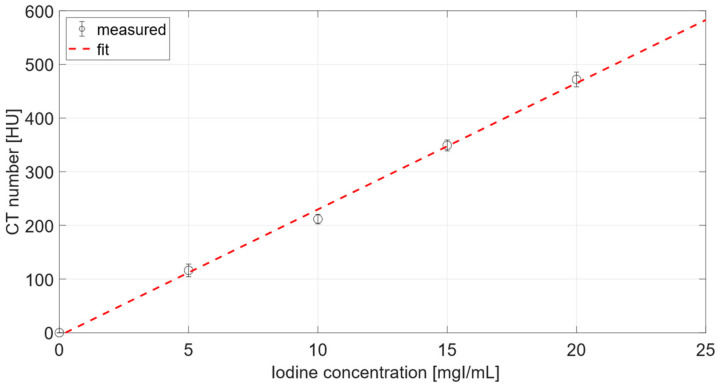
Calibration curve relating CT numbers and iodine concentration of CA4+ from CT reconstructions.

**Figure 6 diagnostics-12-02111-f006:**
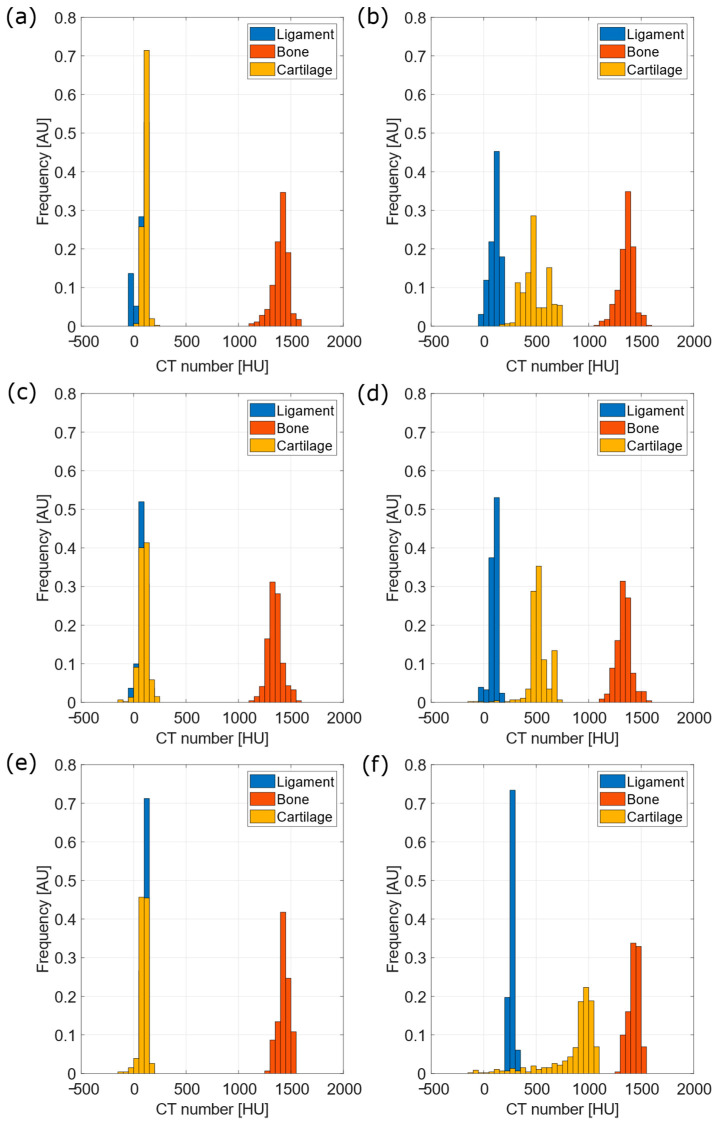
Histograms of CT number distribution, from ROIs drawn on tibiae acquisitions, distinguishing three different tissue types, namely connective, bone and cartilage. On the left, the distributions of CT numbers prior to CA4+ immersion are reported (**a**,**c**,**e**). On the right, the distributions of CT numbers after the 24-h CA4+ immersion are reported (**b**,**d**,**f**). From the top, 5 mgI/mL, 10 mgI/mL and 15 mgI/mL-concentration data are displayed.

**Figure 7 diagnostics-12-02111-f007:**
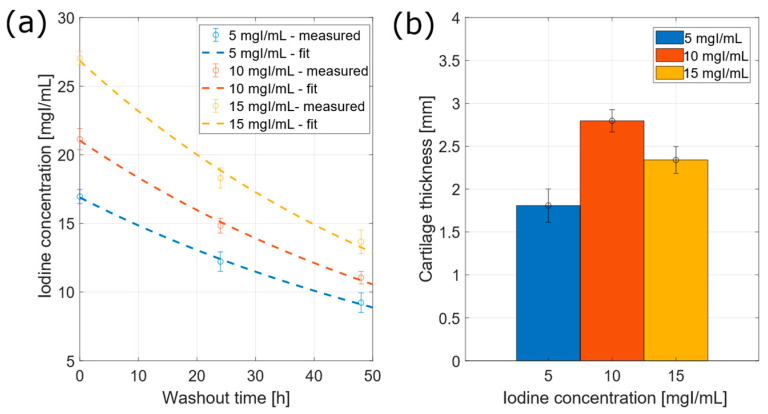
Results following PBS washout of bovine tibiae. (**a**) Washout curve of CA4+ from cartilage to PBS bath, for different concentrations, pointing out the iodine concentration retained in cartilage with respect to washout time. (**b**) Cartilage thickness reported for each tibia, after 24-h bath in CA4+ solution at different iodine concentration.

**Table 1 diagnostics-12-02111-t001:** Results from statistical analysis, including Cohen’s d, Kolmogorov-Smirnov and Mann-Whitney test, obtained from tibiae prior to CA4+ immersion (L = connective, B = bone, C = cartilage).

	5 mgI/mL			10 mgI/mL			15 mgI/mL		
	Cohen’s d	KS Test	MW Test	Cohen’s d	KS Test	MW Test	Cohen’s d	KS Test	MW Test
L-B	21.4	*p* < 0.001	*p* < 0.001	22.4	*p* < 0.001	*p* < 0.001	33.1	*p* < 0.001	*p* < 0.001
B-C	24.0	*p* < 0.001	*p* < 0.001	21.4	*p* < 0.001	*p* < 0.001	30.1	*p* < 0.001	*p* < 0.001
C-L	0.78	*p* < 0.001	*p* < 0.001	0.22	0.313	0.879	0.33	*p* < 0.001	*p* < 0.001

**Table 2 diagnostics-12-02111-t002:** Results from statistical analysis, including Cohen’s d, Kolmogorov–Smirnov and Mann–Whitney test, obtained from tibiae after CA4+ immersion (L = connective, B = bone, C = cartilage).

	5 mgI/mL			10 mgI/mL			15 mgI/mL		
	Cohen’s d	KS Test	MW Test	Cohen’s d	KS Test	MW Test	Cohen’s d	KS Test	MW Test
L-B	19.5	*p* < 0.001	*p* < 0.001	21.6	*p* < 0.001	*p* < 0.001	28.6	*p* < 0.001	*p* < 0.001
B-C	8.7	*p* < 0.001	*p* < 0.001	9.5	*p* < 0.001	*p* < 0.001	3.3	*p* < 0.001	*p* < 0.001
C-L	4.3	*p* < 0.001	*p* < 0.001	5.9	*p* < 0.001	*p* < 0.001	3.5	*p* < 0.001	*p* < 0.001

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
