# Peer review of "A Cationic Contrast Agent in X-ray Imaging of Articular Cartilage: Pre-Clinical Evaluation of Diffusion and Attenuation Properties"

_diagnostics, 2022, doi:10.3390/diagnostics12092111_

Round 1

Reviewer 1 Report

In the study of Fantoni et al., diffusion of a cationic contrast agent CA4+ (originally developed by Prof. Grinstaff from Boston University) is studied using a micro-CT and a clinical scanners. First of all, The title is misleading. I don't think 'novel' applies anymore to CA4+ as it has been in use for over a decade. The abstract is not successful in presenting what was done. 'Millimeter and micrometer' scales don't allow to see that two different experiments were conducted. I also was struggling to understand this from the Methods. Perhaps a table of workflow of the two would better allow comparison. Ultimately, the results are presented so that you cannot even compare the two results. 

Did it really take 12 people to make this study? Please state any impact or novelty in the text. At the moment there is none. CA4+'s ability to enhance contrast has been shown in dozens of previous studies. Any kind of reference method would allow at least validating the results.

Introduction

-Authors have familiarized themselves well with the existing literature. 

Methods

-I recommend rewriting the methods. I was struggling to follow the work order. I suggest adding a work flow figure. Reader is not given enough information so that they could understand the micrometer scale samples. 

Chapter 2.1.2 

-As it is known to alter the tissue, What was the osmolality of the contrast agent solutions?

Chapter 2.1.4. 

-I'm struggling to understand what is the purpose of this experiment?

-Title is missing apostrophe. I recommend rewording.

-"Afterwards, PBS was removed"? From the samples?

-Did the 0.5ml of the CA4+ stay on top of the samples? What was the size of the sample holder? Estimating that the cartilage thicknesses were around 2mm, average sample volume would have been around 157ml. CA4+ concentration inside cartilage reaches several times the concentration what it is in the bath (dependent on the concentration). So your relative concentration inside a sample compared to the bath might have been very very low. 

Chapter 2.1.5.

-Version number of Matlab?

-Why not just report the characteristic diffusion times and be better able to compare your results to previous works?

-What was the size of the rectangular ROI?

Chapter 3.1.1

-What is unit GL? Why not use Housfields? I don't understand how much this is. Especially if you used only 0.5ml of contrast agent per sample.

Chapter 3.1.2

-Figure 1: "Microradiograph of osteochondral plug after 24h immersion". And this is now with the 0.5ml droplet? Please name in the figure the "air, CA4+ bath, articular cartilage, and subchondral bone". I recommend separating the a and b figures as they are not related. Why the macroradiographs (millimeter scale) images are not presented?

Chapter 2.2.2

-What's the difference between the samples of 2.1.4 and 2.2.2? What are the dimensions here? A photograph of the samples would clarify.

-Can authors speculate how the dilution of the bath due to diffusion has altered the results?

-Was the PBS changed during the washout?

Chapter 2.2.3.

-What was the size of the ROIs?

Figure 2: Please change the order in your text boxes as the order inside the figure is reversed. 

Figure 4: How do these results compare against other calibration curves, for instance Bhattarai et al. [http://link.springer.com/10.1007/s10439-018-2013-y]?

Figure 5: What was the immersion time here again? Please consider using box plots instead of histograms. 

Figure 6b: "Cartilage thickness reported for each tibia". What about Femurs? How has the significantly different thickness altered the concentration and more importantly the agent saturation time? 

Chapter 5

-Very ambiguous and not quite scientific: "The preliminary results of PBS washout point out the possible CA4+ removal from articular cartilage". What do you mean by 'possible'? You studied this. 

Author Response

Please consider the responses to your comments included in the manuscript in red.

In the study of Fantoni et al., diffusion of a cationic contrast agent CA4+ (originally developed by Prof. Grinstaff from Boston University) is studied using a micro-CT and a clinical scanners. First of all, The title is misleading. I don't think 'novel' applies anymore to CA4+ as it has been in use for over a decade. The abstract is not successful in presenting what was done. 'Millimeter and micrometer' scales don't allow to see that two different experiments were conducted. I also was struggling to understand this from the Methods. Perhaps a table of workflow of the two would better allow comparison.

Authors: Thank you for your consideration about the title. As you recommended, we excluded the term “novel”. For the sake of clarity we also replaced “Millimeter scale” with “Clinical CT study”, and “Micrometer scale” with “microCT study”.

Please note the different purpose for the microCT and the clinical CT assessments. MicroCT analysis allowed us to obtain information about the diffusion times and the depth-wise distribution of CA4+ at micrometer scale. The saturation time was then used to validate the times considered for the subsequent immersion of tibiae in CA4+ solution, for the clinical CT acquisitions. Clinical CT acquisitions highlighted the outcome of CA4+ on tibial plates, delivered with instrumentation regularly employed in clinical routine. In particular, the resulting three-dimensional iodine map at millimetric scale pointed out the differential CA4+ distribution, which in turn depends on cartilage’s local features. Since the animal model chosen for the experiment (namely, the bovine model) approximates very well the human tibial plate, the clinical CT measurements anticipate the possible CA4+ outcome on human knee condyles.

Please find the above-mentioned considerations at lines 86-95.

Ultimately, the results are presented so that you cannot even compare the two results. 

Authors: Due to the different nature of the assessments (namely, two-dimensional acquisitions for microCT and three-dimensional acquisitions for the clinical CT), indeed carried out at different experimental conditions, it is unpractical to carry out any comparison between them. Owing to the significant difference between the spatial resolution of the X-ray imaging systems used in our study (approximately 11.5 µm for microCT vs. 0.625 mm-slice thickness for the clinical CT), it is not possible to carry out any comparison between the depth-wise CA4+ distribution reached in cartilage.

Rather than a comparison, the present study is based on two experiments delivering complementary results.

At least, accordance is found on iodine concentration in cartilage, between the clinical measurements on tibia immersed in 10 mgI/ml-concentration solution and microCT assessments on tibial osteochondral plugs, as you can find at lines 470-476 of Discussion chapter, and below:

“It is noteworthy that there is accordance between the iodine concentration in tibial osteochondral plugs  (measured with microCT) and tibiae (measured with the clinical CT. As a matter of fact, microCT assessment pointed out that nearly the same attenuation was reached between 6 h and 24 h. Hence the bath time adopted for tibiae was adequate, though lower temperatures ruled out CA4+ diffusion.  Furthermore, the different sample preparation adopted for microCT and clinical CT acquisitions respectively seemingly did not impact on CA4+ final uptake.”

Did it really take 12 people to make this study?

Authors: This is a multidisciplinary work, which involved different research units. Please find the author contributions below, listed in the final pages of the study:

Author Contributions: Conceptualization, S.F., I.G., P.C., M.A., A.Z.B., D.C. and F.B.; methodology, S.F., I.G., P.C., M.A., A.Z.B., D.C. and F.B.; software, S.F. and I.G.; validation, S.F. and I.G..; formal analysis, S.F. and I.G..; investigation, S.F., I.G., M.A., A.Z.B., D.C. and F.B.; resources, P.C., A.T, V.C., C.T. and F.B.; data curation, S.F. and I.G.; writing—original draft preparation, S.F.; writing—review and editing, S.F., I.G., P.C., G.P., A.T., V.C., C.T., A.B., I.G., M.A., A.Z.B., D.C. and D.C.; visualization, S.F. and I.G.; supervision, ; project administration, M.A., A.Z.B., D.C. and F.B.; funding acquisition, P.C., A.T., A.B. and F.B. All authors have read and agreed to the published version of the manuscript.

Please state any impact or novelty in the text. At the moment there is none. CA4+'s ability to enhance contrast has been shown in dozens of previous studies. Any kind of reference method would allow at least validating the results.

Authors: From the previous studies, no evidence of complete tibial condyle acquisition with clinical CT emerges. If studies including intact joints are considered, assessments were carried out with in vivo microCT, as the animal model used (namely rabbit, rat) cannot be investigated with any clinical CT system. Furthermore, those models do not replicate in size the human knee joint.

The only work focusing on whole stifle joint is from Stewart et al.:

  1. Stewart, R.C.; Nelson, B.B.; Kawcak, C.E.; Freedman, J.D.; Snyder, B.D.; Goodrich, L.R.; Grinstaff, M.W. Contrast-Enhanced Computed Tomography Scoring System for Distinguishing Early Osteoarthritis Disease States: A Feasibility Study. J Orthop Res 2019, 37, 2138–2148, doi:10.1002/jor.24382.

The above-mentioned study investigated the equine femoropatellar joint and how the CA4+ injection highlighted cartilage defects, using a clinical CT system. However, no attention was drawn to tibial cartilage, though its deterioration, together with the femoral cartilage, is implied in the development of osteoarthritis.

The present work aims to fill the gap, by considering the outcome of CA4+-induced contrast enhancement on whole tibial plates, by means of a clinical CT system.

Furthermore, the present study focused on the positive impact of CA4+ use on differentiation between three different tissues (namely, connective tissue, cartilage and bone), clinically relevant for evaluation of joint state. Please find these considerations in Discussion section, at line 369-383.

Introduction

-Authors have familiarized themselves well with the existing literature. 

Authors: Thank you for the encouraging comment.

Methods

-I recommend rewriting the methods. I was struggling to follow the work order. I suggest adding a work flow figure. Reader is not given enough information so that they could understand the micrometer scale samples. 

 Authors: Please find the information included at lines 86-95.

Chapter 2.1.2 

-As it is known to alter the tissue, What was the osmolality of the contrast agent solutions?

Authors: As no measurement of osmolality of CA4+ solution was carried out, it is one of the limitations of this study. Please consider the following paragraph, included in the Discussion chapter, at lines 504-511, with the relative citations supporting the reported evidences:

“Another shortcoming of the present work is the lack of control over the osmolality of the obtained CA4+ solution. Any difference in osmolality between the internal and external cartilage environment might affect diffusion and morphological properties, though slight modifications were observed on equine samples with non-ionic, small molecule contrast agent [48]. Contrarywise, the use of hyperosmolar contrast agents heavily impacts on cartilage mechanical properties, causing tissue softening and shrinkage [49]. As a consequence, possible effects of CA4+ solution osmolality should be taken into account in the frame of conjoint biomechanical investigations.”

  1. Pouran, B.; Arbabi, V.; Zadpoor, A. A.; Weinans, H. Isolated Effects of External Bath Osmolality, Solute Concentration, and Electrical Charge on Solute Transport across Articular Cartilage. Medical Engineering & Physics 2016, 38 (12), 1399–1407. https://doi.org/10.1016/j.medengphy.2016.09.003.
  2. Turunen, M. J.; Töyräs, J.; Lammi, M. J.; Jurvelin, J. S.; Korhonen, R. K. Hyperosmolaric Contrast Agents in Cartilage Tomography May Expose Cartilage to Overload-Induced Cell Death. Journal of Biomechanics 2012, 45 (3), 497–503. https://doi.org/10.1016/j.jbiomech.2011.11.049.

Chapter 2.1.4. 

-I'm struggling to understand what is the purpose of this experiment?

Authors: The aim of the microCT acquisitions is the assessment of the saturation time, and the evaluation of the depth-wise distribution of CA4+ in cartilage at micrometric resolution (11.5 µm pixel size). In fact, the resolution of the clinical CT reconstructions (pixel size 0.32mm x 0.32mm x 0.625mm) is inadequate to assess the depth-wise distribution of CA4+.

Please find the rephrased statement at lines 79-85 of Introduction chapter.

-Title is missing apostrophe. I recommend rewording.

Authors: Please check the new title for the subsection 2.1.4. “MicroCT acquisition of osteochondral plugs”

-"Afterwards, PBS was removed"? From the samples?

-Did the 0.5ml of the CA4+ stay on top of the samples? What was the size of the sample holder?

Authors:

The custom-made PMMA sample-holder is shown in the figure below:

The sample-holder features an external diameter of 12 mm, and an inner diameter of 10 mm. Therefore, the osteochondral plug could easily be inserted. The container can host osteochondral plugs with variable height, and the contrast agent bath at the top the cartilage surface. Provided that the height of the sample-holder interior is equal to approximately 3 cm, a total of 3 ml of volume is admitted inside the sample-holder, including the osteochondral plug and the CA4+ bath.

Please find the description at lines 135-140.

The procedure for microCT acquisition of osteochondral plugs included the following steps:

 - prior to the insertion of the osteochondral plugs in the sample-holder, the sample-holder inner walls were smeared with silicone-based grease, to prevent any leakage of CA4+ through the lateral edges of articular cartilage, once the osteochondral plug is inserted. Note that the inner diameter of the sample-holder approaches the diameter of the osteochondral plugs, so the amount of silicone-based grease was minimal.

 - the osteochondral plug was inserted in the sample-holder, directing the cortical bone to the bottom of the sample holder. In this configuration, the articular cartilage is directed upwards, and can easily face the bath of contrast agent.

 - over the cartilage surface was poured 0.5ml of PBS, and one planar “baseline” acquisition was taken.

 - after the baseline acquisition, the PBS on the top of the articular cartilage was removed from the sample-holder, and replaced with 0.5ml of CA4+.      

 -  planar acquisitions were taken at different time points (namely 2, 5, 10, 15, 20, 30, 45 minutes, 1, 1.5, 2, 3, 4, 5, 6, 24, 25, 26 hours), keeping the osteochondral plus in CA4+ bath.

Find the above-mentioned description at lines 141-153.           

 Estimating that the cartilage thicknesses were around 2mm, average sample volume would have been around 157ml. CA4+ concentration inside cartilage reaches several times the concentration what it is in the bath (dependent on the concentration). So your relative concentration inside a sample compared to the bath might have been very very low. 

Authors: There might be a misunderstanding. The average cartilage volume for an osteochondral plug is estimated to be about 0.157 mm3, which corresponds to 0.157 ml. As a result, the CA4+ volume is approximately 3 times the cartilage volume. Indeed, the results of the present work confirmed that the considered amount of CA4+ solution is enough to induce optimal contrast enhancement in cartilage, as the cartilage attenuation reaches approximately 7 times the attenuation of CA4+ bath.

Please find the statement at lines 150-153.

Chapter 2.1.5.

-Version number of Matlab?

Authors: Matlab 2021a. Please find the missing information at line 158.

-Why not just report the characteristic diffusion times and be better able to compare your results to previous works?

Authors: We were interested to assess the time at which diffusion of CA4+ reaches equilibrium inside the cartilage. In anticipation to further preclinical tests, the saturation time is more appealing for clinicians, rather than the characteristic time, by definition the time required for cartilage to reach 65% of maximum attenuation. However, the conversion from saturation time τ95 to the characteristic time τ is immediate, as pointed out by the following formula:

We included the relationship between characteristic and saturation time in Material and methods chapter, at lines 166-170.

-What was the size of the rectangular ROI?

 Authors: The ROI was 100 pixel wide (equivalently, 1.15 mm), while its height was adjusted to cartilage thickness. Nothing but articular cartilage was included in the ROI. Please find this information included at lines 175-176.

Chapter 3.1.1

-What is unit GL? Why not use Housfields? I don't understand how much this is. Especially if you used only 0.5ml of contrast agent per sample.

Authors: Please note that for planar acquisitions it is not possible to consider CT numbers, as no reconstruction is performed (in other words, no attenuation coefficient µ is retrieved). GL (Gray Levels) is the attenuation value provided directly by the microCT system. From the calibration curve, the conversion from GL to mgI/ml allowed us to evaluate the actual iodine concentration dispersed in cartilage.

Chapter 3.1.2

-Figure 1: "Microradiograph of osteochondral plug after 24h immersion". And this is now with the 0.5ml droplet? Please name in the figure the "air, CA4+ bath, articular cartilage, and subchondral bone". I recommend separating the a and b figures as they are not related. Why the macroradiographs (millimeter scale) images are not presented?

Authors: Please find the updated figure 1, and the calibration curve from microCT assessment in Figure 2. Macroradiographs were not included for the sake of manuscript compactness.

Chapter 2.2.2

-What's the difference between the samples of 2.1.4 and 2.2.2? What are the dimensions here? A photograph of the samples would clarify.

Authors: In Subsection 2.1.4. is illustrated the preparation of osteochondral plugs. The osteochondral plugs are cylinders including different tissues (cartilage tissue, subchondral bone, trabecular bone), of 10 mm-diameter and variable height (average 2-3 cm). The surface of the articular cartilage of the resulting osteochondral plugs is about 0.78 cm2. The osteochondral plugs were obtained from both femurs and tibiae distal ends of two bovine stifle joints. Please find the specified size of osteochondral plugs at lines 95-97.

The samples examined in Subsection 2.2.2. were the distal ends of tibiae only, extracted from three bovine stifles. The size of the tibial plates are considerably larger (about 15 cm x 15 cm), as their articulating surface reaches 225 cm2. Please find the specified size of tibial plates at lines 197-198.

-Can authors speculate how the dilution of the bath due to diffusion has altered the results?

Authors: In the definition of Materials and Methods, our expectations were defined by previous literature. For example, in their work Karhula et al. used different concentrations of CA4+:

  1. Karhula, S.S.; Finnilä, M.A.; Freedman, J.D.; Kauppinen, S.; Valkealahti, M.; Lehenkari, P.; Pritzker, K.P.H.; Nieminen, H.J.; Snyder, B.D.; Grinstaff, M.W.; et al. Micro-Scale Distribution of CA4+ in Ex Vivo Human Articular Cartilage Detected with Contrast-Enhanced Micro-Computed Tomography Imaging. Frontiers in Physics 2017, 5.

namely 3, 6 and 24 mgI/ml. It should be noted that the samples included were not bovine but human. Furthermore, the dimensions and the number of samples (2 mm-diameter, n=9 osteochondral plugs from patients femurs, 3 for each iodine concentration) considered differ from our study (10 mm-diameter, n=9 osteochondral plugs from bovine tibial distal ends, n=9 osteochondral plugs from bovine femur distal end).

The work of Karhula et al. evidenced an increase in cartilage attenuation for increasing iodine concentrations, as it is possible to observe from “whole sample” data in Figure 3 of the same work.

Despite the different nature of samples from our study, we referred to such evidence for the design of the clinical CT experiment.

Please find our statement about the definition of CA4+ concentrations at lines 208-209, and the relative discussion at lines 440-446.

-Was the PBS changed during the washout?

Authors: Yes, the PBS bath was changed after each washout procedure. Please find the included information at lines 213-214.

Chapter 2.2.3.

-What was the size of the ROIs?

Authors: The ROIs were circular, of 5 pixels-radius. Given the pixel size of 0.32 mm, the ROIs had radius 1.6 mm. Please find the information included at lines 224-225.

Figure 2: Please change the order in your text boxes as the order inside the figure is reversed. 

Authors: Please find the updated figure (now Figure 3) in the manuscript.

Figure 4: How do these results compare against other calibration curves, for instance Bhattarai et al. [http://link.springer.com/10.1007/s10439-018-2013-y]?

Authors: Considering the curve in Figure 2b from the work of Bhattarai at al., and the calibration curves from microCT (Figure 2) and clinical CT (Figure 5) assessments, both curves are definitely linear. However, it is not possible to carry out further comparison as the units and scales displayed on y-axis of the curve of Bhattarai et al. are different from the one of the present work.

Please find the aforementioned consideration at lines 384-388.

Figure 5: What was the immersion time here again? Please consider using box plots instead of histograms. 

Authors: The immersion time is 24 hours. Please find the modified caption below the figure (now Figure 6) with the added information at line 325. We considered the histograms rather than the box plot, as the former gives a direct insight into the considerable distinction between tissues, before and after the CA4+ immersion, especially in the case of connective tissue and articular cartilage.

Figure 6b: "Cartilage thickness reported for each tibia". What about Femurs? How has the significantly different thickness altered the concentration and more importantly the agent saturation time? 

Authors: We apologize, as the number and nature of samples considered for the clinical CT acquisition was specified in the Abstract, but not in Subsection 2.2.2. of the Materials and Methods. Please find the Subsection 2.2.2. included with the missing information.

For the clinical CT acquisitions, only the distal ends of tibiae were considered. In particular, three tibiae were used, and no femur was included for these assessments.

The impact of different thicknesses on resulting CA4+ concentration in cartilage was already shown in the microCT results, at lines 418-422. In fact, slight difference is found in cartilage attenuation (and in turn even in iodine concentration) after 24 hours of immersion, as reported in Figure 4. Together with the difference between the saturation time evaluated for femoral and tibial osteochondral plugs (namely, 42 minutes), those slight discrepancies reflect the as much slight different morphology and compositional. Due to the planar nature of microCT acquisitions, we preferred not to report the cartilage thicknesses in the work, as their reliability is not comparable to thickness measurements carried out with 3D reconstructions. However, rough measurements from planar images highlighted mean thicknesses of 1.15 ± 0.36 mm and 1.30 ± 0.62 mm, for femoral and tibial cartilage of osteochondral plugs, respectively. Not only the tibial cartilage is thicker, but its measurements are subject to higher fluctuations (namely, the standard deviation of tibial cartilage thickness is greater than the femoral cartilage).

The search for any relationship between cartilage thickness and X-ray attenuation is beyond the aims of this work, given the entity relatively small of the discrepancies between the tibial and femoral cartilage. The magnitude of the fluctuations found in attenuation values (referring to error bars in Figure 4) depends on the number of samples included in the study. A greater number of samples will settle such fluctuations to lower values.

Chapter 5

-Very ambiguous and not quite scientific: "The preliminary results of PBS washout point out the possible CA4+ removal from articular cartilage". What do you mean by 'possible'? You studied this. 

Authors: Thank you for the suggestion. Effectively, we made a vague statement. Please find the quote rephrased at line 533, and below:

"The preliminary results of PBS washout point out the partial CA4+ removal from articular cartilage".

Reviewer 2 Report

Your paper is well written and reproduces other's report. Please make any comparison with the paper in Sci Rep 2019; 9: 7118.  It seems better to demonstrate images of three layers of normal and degenerative cartilages. Could you also see the calcified layer? What is the explanation of different saturation time of femur and tibia? Is that related to amount of PG and/or water?

I wonder how this microCT applys to human studies and how the treatment of osteoarthritis can be improved with early detection of osteoarthritis. 

Page 2, line 2-3: Magnetic Resonance ----->magnetic resonance

                           Computed Tomography------>computed tomography

Author Response

Please consider the responses to your comments included in the manuscript in green.

Your paper is well written and reproduces other's report. Please make any comparison with the paper in Sci Rep 2019; 9: 7118.  It seems better to demonstrate images of three layers of normal and degenerative cartilages.

Authors: Thank you for the consideration.

We agree with you on your statement, concerning the comparison of our results with the work of Saukko et al.:

  1. Saukko, A.E.A.; Turunen, M.J.; Honkanen, M.K.M.; Lovric, G.; Tiitu, V.; Honkanen, J.T.J.; Grinstaff, M.W.; Jurvelin, J.S.; Töyräs, J. Simultaneous Quantitation of Cationic and Non-Ionic Contrast Agents in Articular Cartilage Using Synchrotron MicroCT Imaging. Scientific Reports 2019, 9, 7118, doi:10.1038/s41598-019-43276-6.

The three-dimensional acquisition is certainly more accurate than the two-dimensional (planar). We discussed the benefit (namely, reduced acquisition times) and drawback (namely, overlap of structures from different tissues in projection) of the latter, in the Discussion Chapter:

The choice of acquiring fast planar images avoids any bias introduced by the ongoing diffusion process of CA4+ within cartilage, potentially occurring during tomographic acquisitions. On the other hand, only 2-dimensional information is provided. Due to the heterogeneous nature of samples, image projections may include tissue regions with different local properties. Image acquisition should be achieved avoiding any overlap between cartilage and subchondral bone, cartilage and CA4+ bath.

Please find the statement, reporting the comparison to the work of Saukko et al. at lines 458-463 of the Discussion chapter.

Could you also see the calcified layer?

Authors: Accordingly to the statement reported above, we were not able to distinguish the calcified layer. It is another drawback of planar acquisition. Please find the statement in Discussion Chapter, at line 458.

What is the explanation of different saturation time of femur and tibia? Is that related to amount of PG and/or water?

Authors: Such aspect was partially discussed in Discussion Chapter (lines 415-425), as reported below:

The fluctuations associated to the attenuation values shown in the diffusion curve are attributable to the heterogeneous local properties of cartilage. In particular, morphological (i.e. thickness) and compositional (i.e. proteoglycans content) features are expected to vary, depending on the extraction site on articulating cartilage [42]. Such fluctuations, highlighted by the error bars in Figure 4, impacted on the parameters of the fitting function, and, in turn, on the evaluation of the saturation time. As a result, a discrepancy of 42 minutes was found between the saturation time determined for femoral and tibial osteochondral plugs. On the other hand, the magnitude of the fluctuations found in attenuation values (referring to error bars in Figure 4) depended on the number of samples included in the study. A greater number of samples should settle such fluctuations to lower values. These remarks are confirmed by the clinical CT assessments. As a matter of fact, the VOIs provide different cartilage thickness, as reported in Figure 6, with fluctuations from mean value ranging from 5% to 10%. Different thickness values are attributable to variations of CA4+ permeation in cartilage. Furthermore, the immersion operation of the tibia distal end in silicon sheath is user-dependent.

The impact of different thicknesses on resulting CA4+ concentration in cartilage was shown in the microCT results. In fact, a slight difference is found in cartilage attenuation (and in turn in iodine concentration) after 24 hours of immersion, as reported in Figure 3. Together with the difference between the saturation time evaluated for femoral and tibial osteochondral plugs (namely, 42 minutes), this discrepancy should reflect the slight different morphology (namely, thickness) and composition (namely, proteoglycan concentration, collagen distribution and water content) of the samples.

The different thickness between tibial and femoral articular cartilage gives some hint about the discrepancy observed in saturation times. Due to the planar nature of our microCT acquisitions, we preferred not to report the cartilage thicknesses, as their reliability is not comparable to thickness measurements carried out with 3D reconstructions. However, rough measurements from planar images highlighted mean thicknesses of 1.15 ± 0.36 mm and 1.30 ± 0.62 mm, for femoral and tibial cartilage of osteochondral plugs, respectively. Not only the tibial cartilage is thicker, but its measurements are subject to higher fluctuations (namely, the standard deviation of tibial cartilage thickness is greater than the femoral cartilage).

The search for any relationship between cartilage thickness and X-ray attenuation is beyond the aims of this work, given the relatively small discrepancies between the tibial and femoral cartilage. The magnitude of the fluctuations found in attenuation values (referring to error bars in Figure 3) depends on the number of samples included in the study. A greater number of samples should settle such fluctuations to lower values.

Furthermore, any relationship between cartilage thickness solely and saturation time might be misleading. In fact, the articular cartilage denotes both morphological and compositional features, which must be disentangled. The adoption of reference methods, like MRI, histology and cartilage mechanical tests, should shed light on sample composition and clarify the relationship between X-ray attenuation and proteoglycan content.

Please find the statement at lines 418-425 of Discussion chapter.

I wonder how this microCT applys to human studies and how the treatment of osteoarthritis can be improved with early detection of osteoarthritis.

Authors: Please find the response to your comment in the Conclusions chapter, at lines 539-557 of Conclusions chapter.

Assumed the compatibility of bovine model to human in osteoarthritis studies [51], it is possible to replicate the microCT study on ex vivo human osteochondral plugs. Recent microCT systems, featuring high resolution acquisitions and reduced scan times, should allow three-dimensional assessments besides planar acquisitions.

Obviously, ex vivo experiments are destructive, as the extraction of sample requires explants from cadavers or arthroplasties. The samples must be suitable for the microCT, thus their size tends to be small. Furthermore, samples harvested from arthroplasties’ explants might already show development of osteoarthritis, even at late stages.

On the other hand, those drawbacks can be avoided if pre-clinical microCT systems are adopted. In particular, pre-clinical microCT systems perform high resolution peripheral quantitative CT (HRpQCT) in vivo acquisitions on distal ends of patients’ limbs [52–54]. Not only it is possible to carry out in vivo assessment, but the dose deposited to patients is intrinsically reduced if compared to conventional CT, as reported by recent literature [9]. Provided the approval by FDA and the design of an optimal protocol for contrast agent usage in patients, the application of contrast agents should enable the contrast enhancement of articular cartilage. As a consequence, the state of both articular cartilage and subchondral bone would be unveiled, and possible staging of osteoarthritis would be performed.

  1. Cope, P.J.; Ourradi, K.; Li, Y.; Sharif, M. Models of Osteoarthritis: The Good, the Bad and the Promising. Osteoarthritis Cartilage 2019, 27, 230–239, doi:10.1016/j.joca.2018.09.016.
  2. Whittier, D.E.; Boyd, S.K.; Burghardt, A.J.; Paccou, J.; Ghasem-Zadeh, A.; Chapurlat, R.; Engelke, K.; Bouxsein, M.L. Guidelines for the Assessment of Bone Density and Microarchitecture in Vivo Using High-Resolution Peripheral Quantitative Computed Tomography. Osteoporos. Int. 2020, 31, 1607–1627, doi:10.1007/s00198-020-05438-5.
  3. Keen, C.E.; Whittier, D.E.; Firminger, C.R.; Edwards, W.B.; Boyd, S.K. Validation of Bone Density and Microarchitecture Measurements of the Load-Bearing Femur in the Human Knee Obtained Using In Vivo HR-PQCT Protocol. J. Clin. Densitom. Off. J. Int. Soc. Clin. Densitom. 2021, 24, 651–657, doi:10.1016/j.jocd.2021.01.004.
  4. Mys, K.; Varga, P.; Stockmans, F.; Gueorguiev, B.; Neumann, V.; Vanovermeire, O.; Wyers, C.E.; van den Bergh, J.P.W.; van Lenthe, G.H. High-Resolution Cone-Beam Computed Tomography Is a Fast and Promising Technique to Quantify Bone Microstructure and Mechanics of the Distal Radius. Calcif. Tissue Int. 2021, 108, 314–323, doi:10.1007/s00223-020-00773-5.

Page 2, line 2-3: Magnetic Resonance ----->magnetic resonance

                           Computed Tomography------>computed tomography

Authors: Please find the retyped terms in the manuscript.

Round 2

Reviewer 1 Report

I'd like to thank the Authors for their work in revising the manuscript. All my comments were satisfactorily addressed. Replacing the terminology to 'clinical CT' and 'microCT' has improved the clarity. The only last recommendation I have is to add the acronym 'GL' in the text (L.129) to clarify the gray level unit to the readers.